# The Long-Term DEHP Exposure Confers Multidrug Resistance of Triple-Negative Breast Cancer Cells through ABC Transporters and Intracellular ROS

**DOI:** 10.3390/antiox10060949

**Published:** 2021-06-11

**Authors:** Mahendra Jadhao, Eing-Mei Tsai, Ho-Chun Yang, Yih-Fung Chen, Shih-Shin Liang, Tsu-Nai Wang, Yen-Ni Teng, Hurng-Wern Huang, Li-Fang Wang, Chien-Chih Chiu

**Affiliations:** 1Department of Medicinal and Applied Chemistry, Kaohsiung Medical University, Kaohsiung 807, Taiwan; mah.jadhao@yahoo.com or; 2Department of Obstetrics and Gynecology, Kaohsiung Medical University Hospital, Kaohsiung 807, Taiwan; tsaieing@kmu.edu.tw; 3The Graduate Institute of Medicine, College of Medicine, Kaohsiung Medical University, Kaohsiung 807, Taiwan; 4Department of Biotechnology, Kaohsiung Medical University, Kaohsiung 807, Taiwan; as0192071@gate.sinica.edu.tw (H.-C.Y.); liang0615@kmu.edu.tw (S.-S.L.); 5Institute of Biomedical Science, National Sun Yat-sen University, Kaohsiung 804, Taiwan; sting@mail.nsysu.edu.tw; 6Graduate Institute of Natural Products, College of Pharmacy, Kaohsiung Medical University, Kaohsiung 807, Taiwan; yihfungchen@kmu.edu.tw; 7Department of Public Health, College of Health Science, Kaohsiung Medical University, Kaohsiung 807, Taiwan; wangtn@kmu.edu.tw; 8Department of Biological Sciences and Technology, National University of Tainan, Tainan 700, Taiwan; tengyenni@mail.nutn.edu.tw; 9Center for Cancer Research, Kaohsiung Medical University Hospital, Kaohsiung Medical University, Kaohsiung 807, Taiwan; 10Department of Biological Sciences, National Sun Yat-Sen University, Kaohsiung 804, Taiwan; 11Department of Medical Research, Kaohsiung Medical University Hospital, Kaohsiung 807, Taiwan

**Keywords:** breast cancer, long DEHP exposure, ATP-binding transporter proteins, ROS, drug resistance, tariquidar

## Abstract

The characteristics of phthalates had been thought to be similar to endocrine disruptors, which increases cancer risk. The role of phthalates in acquired drug resistance remains unclear. In this study, we investigated the effect of di-(2-ethylhexyl) phthalate (DEHP) on acquired drug resistance in breast cancer. MCF7 and MDA-MB-231 breast cancer cells were exposed to long-term physiological concentration of DEHP for more than three months. Long-exposure DEHP permanently attenuated the anti-proliferative effect of doxorubicin with estrogen receptor-independent activity even after withdrawal of DEHP. Long term DEHP exposure significantly reduced ROS (O_2_^−^) level in MDA-MB-231 cells while increased in MCF7 cells. ATP-binding cassette (ABC) transporters possess a widely recognized mechanism of drug resistance and are considered a target for drug therapy. Upregulation of ABC family proteins, ABCB-1 and ABCC-1 observed in DEHP-exposed clones compared to doxorubicin-resistant (DoxR) and parental MDA-MB-231 cells. A viability assay showed enhanced multidrug resistance in DEHP-exposed clones against Dox, topotecan, and irinotecan. Inhibition of ABC transporters with tariquidar, enhanced drug cytotoxicity through increased drug accumulation reversing acquired multidrug resistance in MDA-MB-231 breast cancer cells. Tariquidar enhanced Dox cytotoxicity by increasing intracellular ROS production leading to caspase-3 mediated apoptosis. Activation of PI3K/Akt signaling enhanced proliferation and growth of DEHP-exposed MDA-MB-231 cells. Overall, long-term DEHP exposure resulted in acquired multidrug resistance by upregulating ABCB-1 and ABCC1; apart from proliferation PI3K/Akt may be responsible for acquired drug resistance through ABC transporter upregulation. Targeting ABCB1 and ABCC1 with tariquidar may be a promising strategy for reversing the acquired multidrug resistance of triple-negative breast cancer cells.

## 1. Introduction

Phthalates are first-choice plasticizers in the synthesis of plastic products. Esters of ortho-phthalic acid are often added to polyvinyl chloride (PVC) products to induce flexibility and durability. Primarily, phthalates are used in industrial and household products including food packaging, baby products, cosmetics/personal care products, and medicinal devices [1,2,3]. Among phthalates, di-(2-ethylhexyl) phthalate (DEHP) is used on a large scale and has beneficial properties for plastic essentials; however, nonconjugate bonding of DEHP results in leaching due to several factors, such as temperature, time of use, and pH; and subsequently results in environmental contamination [4]. In the general population, exposure to phthalates occurs through the ingestion of contaminated food, inhalation of phthalate particulates in the air, and dermal contact through the environment. High molecular weight phthalate (>250 Da) exposure primarily occurs through contaminated food and water [5]. According to the World Health Organization (WHO), 25 µg/kg body weight is the tolerable daily intake (TDI); however, DEHP contributes most to the hazard quotient in hepatotoxicity (HQhep) in approximately 83.1–98.6% of cases [6,7]. The toxicity of this ortho-phthalic acid derivative, including reprotoxicity, neurotoxicity, carcinogenesis, cardiotoxicity, hepatotoxicity, and nephrotoxicity, has been well evidenced over the past several years [8,9,10,11]. In 2020, the International Agency for Research on Cancer (IARC-WHO) reported breast cancer incidence to be highest among all cancers, with 11.7% of new breast cancer cases worldwide [12]. Breast cancer is the leading cause of death among females, which warrants additional attention. Although advancements in cancer screening and therapy have been achieved, chemotherapeutic drug resistance is still a major problem. Drug resistance mechanisms can be classified as intrinsic and acquired, which involve a variety of molecular mechanisms, such as increased drug excretion, drug inactivation, drug target protein alterations, decreased oxidative stress caused by anticancer drugs, increased DNA repair, and apoptosis inhibition or cell survival signal activation [13,14]. Intrinsic drug resistance refers to the existence of drug resistance regulatory factors in cancer cells before chemotherapy; however, in acquired drug resistance, cancer cells become resistant to drugs through mutations induced by drugs or external environmental factors [15]. The exposure of breast cancer cells to plasticizers, such as DEHP, butyl benzyl phthalate (BBP), and di(n-butyl) phthalate (DBP), inhibits the apoptosis mechanism induced by tamoxifen and increases cell proliferation, suggesting that plasticizers may have an effect on resistance to breast cancer drugs [16].

Cell membranes are believed to be able to regulate the entry and exit of substances from cells. The ATP-binding cassette (ABC) transporters located in the cellular membrane play a crucial role in the efflux of cancer therapy drugs and contribute to chemoresistance [17]. Notably, multidrug resistance protein 1 (MDR1/ABCB1), MDR-associated protein 1 (MRP1/ABCC1), and breast cancer resistance protein (BCRP/ABCG2) confer drug resistance [18]. A clinical study performed in 2006 to evaluate the gene expression of adult acute myeloid leukemia (AML) showed high MDR1/ABCB1 and BCRP/ABCG2 expression in a subset of patients with the worst overall survival and high drug resistance [19]. Increased expression of P-glycoprotein (P-gp)/MRP1 induced acquired drug resistance following exposure to high concentrations of DEHP/MEHP (10 µM) in colon cancer [20]. In sarcoma cells, DEHP increased MDR expression, resulting in decreased anticancer drug cytotoxicity [21]. It is evident that the expression levels of ABC transporters regulate drug accumulation and that ABC transporter overexpression leads to drug resistance in cancer [22]. Targeting ABC transporters, inhibiting their function, and reversing drug resistance is an important clinical strategy to overcome ABC transporter-mediated drug resistance. Tariquidar (XR9576), a third-generation inhibitor featuring a high transporter affinity and a low pharmacokinetic interaction, showed potential P-gp inhibition with no side effects and pharmacokinetic binding [23]. Inhibition of ATPase activity of P-glycoproteins is regulated by Tariquidar, which was also found to overcome P-gp activity in clinical study of patients diagnosed with metastatic adrenocortical cancer [24]. The inhibitory activity of tariquidar is quite promising; as low as 25–80 nM tariquidar can specifically bind with and inhibit the ATPase activity of BCRP/ABCG2 in vivo [25].

Phosphatidylinositol 3-kinase (PI3K)/Akt activation is observed in many cancer types and plays an important role in cell survival and growth [26]. PI3K belongs to the lipid kinase family and is activated by the phosphorylation of phosphatidylinositol (phosphatidylinositol-4,5—biphosphate), lipids in the cell membrane, which further activates Akt signaling. The N-terminal pleckstrin homology (PH) domain of Akt binds with high affinity to phosphatidylinositol-3,4,5-triphosphate (PIP_3_) [27]. Phosphatase and tensin homolog (PTEN), a tumor suppressor that is frequently mutated in different cancer types, plays a crucial role in the regulation of the PI3K/Akt pathway. It encodes phosphatidylinositol-3,4,5-triphosphate 3-phosphatase, which dephosphorylates phosphatidylinositol-3,4,5-triphosphate, preventing PI3K/Akt pathway activation and ultimately suppressing cell growth [28]. Loss of PTEN activates the Akt pathway, as low PTEN levels lead to an increase in phosphatidylinositol-3,4,5-triphosphate, which binds with the Akt PH domain [29]. Enhancement of cell survival through PI3K/Akt results from either activation of antiapoptotic molecules or inhibition of proapoptotic molecules and activation of the MAPK family by Akt. Several studies have reported the association between PI3K and drug resistance in cancer, mainly with ABC transporters. Inhibition of PI3K using wortmannin resulted in inhibition of MRP1-mediated drug efflux; however, no effect was observed on MDR1 efflux in acute myelogenous leukemia [30]. Another study showed that activation of PI3K induced MRP1-mediated multidrug resistance in prostate carcinoma against Dox and paclitaxel [31]. Hyaluronan was found to modulate the regulation of MDR1/P-gp by inhibiting p-Akt activity to overcome multidrug resistance in lymphoma cells [32]. Overall, a strong connection between ABC transporter-mediated drug resistance and PI3K/Akt is possible.

In the current study, we evaluated the effect of long-term exposure to low concentrations (>3 months; 100 nM) of DEHP on MCF7 and MDA-MB-231 breast cancer cells in contrast to most other studies which used short term DEHP exposure at high concentration. The effect of long term DEHP exposure on drug resistance and ROS generation was further evaluated. The role of ABC transporter’s expression in acquired drug resistance was studied following long term DEHP exposure in MDA-MB-231 cells. The effect of ABC transporter inhibitor tariquidar was evaluated to overcome DEHP induced acquired drug resistance. Finally, the underlying mechanism of DEHP exposure mediated acquired drug resistance in triple negative breast cancer was explored using RNA sequencing.

## 2. Materials and Methods

### 2.1. Cell Cultures

The human breast cancer cell lines MDA-MB-231 and MCF7 were selected as the experimental models for this study. Both cell lines were acquired from the American Type Culture Collection (ATCC) and maintained in DMEM (Gibco, Grand Island, NY, USA) with 10% FBS (fetal bovine serum), 0.03% glutamine, the antibiotics penicillin and streptomycin (100 μg/mL), and 1 mM sodium pyruvate. Cells were maintained under humidified conditions containing 5% CO_2_ at 37 °C.

### 2.2. Reagents and Antibodies

DEHP (Sigma, 36735), 98%, was purchased from Sigma-Aldrich (St. Louis, MO, USA) and dissolved in DMSO (Sigma). Dox (Sigma D1515), topotecan (Sigma, 1672257), and irinotecan (Sigma, I1406) were purchased from Sigma-Aldrich, dissolved in DMSO and stored in a freezer at −20 °C at a concentration of 10 mM. Tariquidar (HY-10550) was purchased from MedChemExpress (NJ, USA), dissolved in DMSO (Sigma), and stored in a freezer at −20 °C at a concentration of 10 mM. Thiazolyl blue tetrazolium bromide (MTT) reagent (M5655-500MG) was purchased from Sigma-Aldrich and stored at −20 until use. Antibodies against P-gp/MDR1/ABCB1 (141 kDa, GeneTex, GTX108370), MRP1/ABCC1 (170–220 kDa, Cell Signaling, 14685), α tubulin (51 kDa, GeneTex, GTX618802), GAPDH (35.5 kDa, EMD Millipore, MAB374), SOD2 (24 kDa, Merck, 06-984), PRX (22 kDa, GeneTex, GTX101705), Catalase (65 kDa, Merck, 219010), Caspase 3 (32, 17, and 12 kDa, Cell Signaling, 9662S), poly (ADP-ribose) polymerase-1 (PARP; 116 and 89 kDa, Cell Signaling, 9542S), BAX (20 kDa, Proteintech, 50599-2-IG), BCL2 (26 kDa, Proteintech, 12789-1-AP), AIF (67 kDa, Enzo, ADI-905-168), PI3 Kinase p85 (85 kDa, Cell Signaling, 4257), Phospho-PI3K p85 (Tyr458) (85 kDa, Cell Signaling, 4228), AKT (60 kDa, iReal, IR-171-666), Phospho-AKT (Ser473) (60 kDa, Cell Signaling, 4060S), GSK3β (47 kDa, GeneTex, GTX111192), GSK3β (Ser9) (47 kDa, GeneTex, GTX50090), and Phospho-S6 (Ser235/236) (32 kDa, Cell Signaling, 4858) were used. An Annexin V-FITC apoptosis detection kit was purchased from Strong Biotech Corporation (AVK250) (Taipei, Taiwan).

### 2.3. DEHP Exposure and Stable Clone Formation

In the DEHP exposure group, DEHP was mixed with the culture medium (the final concentration of DEHP was 100 nM), and the cells were exposed for as long as 3 months. After exposure, DEHP was removed and the cells were challenged with Dox (10 nM concentration for 72 h). Dox resistant colonies were selected, amplified, and maintained in culture medium for further investigations (referred to as clones # 1, 2, 3, and 4). After stable DEHP exposed MDA-MB-231 clones formed, no further DEHP exposure was performed.

DoxR MDA-MB-231 cells were established by Dox treatment (10 nM concentration for 72 h). The cells were cultured for 10 days to form colonies. Then, the DoxR colonies were selected and maintained as DoxR cells.

MDA-MB-231 cells without DEHP exposure are further referred to as control cells/untreated cells. Control, DEHP exposed MDA-MB-231, and Dox resistant MDA-MB-231 (DoxR) cells were maintained and cultured with equal passage numbers (10–12 passages) to avoid the effect of senescence.

### 2.4. Colony Formation Assay

Approximately 2 × 10^2^ cells were seeded in 6-well culture plates and cultured in a 5% CO_2_ incubator at 37 °C for 48 h. After incubation, MDA-MB-231 and MCF-7 cells were treated with the indicated concentrations of Dox (0.2, 1, 5, 10 nM) for 72 h. Afterward, the medium was removed, the cells were washed with 1×PBS, and fresh Dox-free medium was replaced every 3 days for 10 days. The cells were fixed with 4% paraformaldehyde and then stained with 1× Giemsa stain (Merck, cat. 1.09204.1000) for 20–30 min, and pictures were taken after washing and drying. The experimental results were analyzed with ImageJ software (ImageJ, NIH, MD, USA).

### 2.5. Evaluation of ROS (O_2_^−^) Formation

Approximately 2.5 × 10^5^ control and long term DEHP exposed MDA-MB-231 and MCF-7 cells were seeded in 6-well culture plates and cultured in a 5% CO_2_ incubator at 37 °C for 24 h. After incubation, cells were treated with Dox (2 µM) for 0, 12, 18, and 24 h. Afterward, the cells were collected and stained with DHE using Muse Oxidative Stress kit (Merck, MHC100111) according to manufacturer protocol. Flow cytometry was performed using Guava Muse Cell Analyzer (Luminex, Austin, TX, USA).

### 2.6. Drug Efflux Assay

Approximately 1 × 10^5^ untreated and DEHP-exposed MDA-MB-231 and MCF7 cells were seeded in a 6-well culture plate. After 16 to 18 h of attachment, 1 μM Dox was added to the wells, and the plate was placed in an incubator for 2 h (simulating drug influx). After the incubation medium was removed, the cells were washed with 1× PBS, fresh medium was added, and the cells were placed in an incubator for 1 h (simulating drug efflux). The cells were suspended, collected using trypsin, washed with 1× PBS, centrifuged at 3000 rpm for 5 min, and resuspended in 200 μL of 1× PBS. The fluorescence intensity of Dox was evaluated using a BD Accuri C6 flow cytometer. The results were analyzed by BD CFlow^®^ software.

### 2.7. Reverse Transcription Polymerase Chain Reaction (RT-PCR)

Total RNA was extracted with Direct-zolTM RNA MiniPrep (Zymo Research, cat. R2052) according to the manufacturer’s instructions. RNA was reverse transcribed to complementary DNA using a High Capacity cDNA Reverse Transcription kit (cat. 4368814, Applied Biosystems, Foster City, CA, USA) following the manufacturer’s instructions.

Quantitative real-time polymerase chain reaction was performed using SYBR Green PCR Master Mix (Applied Biosystems, cat. 4309155) and designed forward and reverse primers, and the temperature and time conditions of the reaction were as follows: 95 °C for 10 min to activate the agent; 40 continuous cycles of 95 °C for 15 s and cooling to 60 °C for 1 min; 95 °C for 15 s, cooling to 60 °C for 1 min, and warming to 95 °C for 15 s to measure the cycle threshold (Ct). The difference in mRNA expression between the experimental group and the control group was calculated by the 2^−^^△△Ct^ equation. The results were analyzed by StepOne Software v2.2.2 (Applied Biosystems, USA) software.

Primers: Accession number, gene name, and primer sequences.


**Accession Number**

**Gene**

**Sequence**
NM_001248946.2ABCB1forward5’-GGGATGGTCAGTGTTGATGGA-3’reverse5’-GCTATCGTGGTGGCAAACAATA-3’NM_004996.4ABCC1forward5’-CAACGGGACTCAGGAGCACA-3’reverse5’-CGGCCATGGAGTAGCCAAAC-3’NM_001101.5β-actinforward5’-TGAGACCTTCAACACCCCAGCCAT-3’reverse5’-CGTAGATGGGCACAGTGTGGGTG-3’

### 2.8. Western Blot Analysis

The western blot assay was performed as described previously [33]. Briefly, the protein concentration was estimated using a Pierce BCA protein estimation kit (cat. 23225, Thermo Fischer, Waltham, MA, USA). Forty micrograms of protein lysate were separated using 10% SDS-polyacrylamide gel electrophoresis (SDS-PAGE) and electrotransferred to PVDF membranes. Afterward, the PVDF membrane was blocked with 5% nonfat milk (Sigma-Aldrich), followed by incubation with primary antibody (diluted according to the antibody datasheet) overnight at 4 °C. After incubation, the membrane was washed with TBST to remove nonspecifically bound primary antibody. Next, the secondary antibodies targeted to the specific primary antibody hosts were added, followed by washing with TBST. The chemiluminescence signals were measured by an ECL^TM^ detection kit (Amersham, Piscataway, NJ, USA) followed by X-ray film exposure in the dark.

### 2.9. Immunofluorescence Staining

Control and DEHP exposed MDA-MB-231 cells (1 × 10^5^) were seeded 12 well plate containing 12 mm cover glass (Deckglaser, Germany) and incubated overnight. Cell were fixed with 4% paraformaldehyde for 10–20 min and permeabilized with 1× PBST for 5 min. Cell blocking was performed in 1% BSA and treated with primary antibody overnight at 4 °C. Cells were washed and incubated with FITC-conjugated secondary antibody for 1 h at room temperature. Cover glasses were mounted on microscope glass slides using mounting medium (Vectashield, Vector Laboratories, CA, USA). Then, confocal imaging was performed using an Olympus FV1000 confocal laser scanning microscope.

### 2.10. Analysis of Dox Resistance by Growth Proliferation Test

Cell growth was analyzed by the trypan blue dye exclusion assay described previously [34]. In brief, the cells were treated with vehicle control (DMSO) or Dox (1 µM) for 24 and 48 h. After incubation, the cells were washed with 1× PBS, trypsinized, exposed to 0.2% trypan blue at a 1:1 ratio and then counted manually using a Neubauer chamber under an inverted light microscope.

### 2.11. In Vivo Zebrafish Xenograft Cell Proliferation

Wild type zebrafish (Danio rerio) were raised and maintained at Zebrafish Crore Facility, KMU. Zebrafish embryos were obtained by pairwise mating. Obtained eggs were incubated in 0.03% Phenylthiourea (PTU) at 28 °C. After 48 h of incubation, embryos were collected. Control MDA-MB-231 and DEHP exposed clone #1 were trypsinized and stained with Vybrant^®^ DiI (Life technologies) according to manufacturer’s protocol. The embryo sac was injected with 10 nL of cell suspension (approximately 300 cells) using a glass needle and a microinjection setup. Embryos were incubated in Dox (25 µM) containing water and imaged after 48 h of injection (48 hpi) using fluorescence microscope. Finally, the tumor size was calculated using formula:Tumor volume (mm^3^) = [(longer diameter) × (shorter diameter)^2^].

### 2.12. Tariquidar Treatment and Drug Resistance Evaluation

Untreated, DEHP-exposed, and DoxR cells (1 × 10^5^) were seeded overnight. The cells were treated with tariquidar (1.5 µM) for 2 h in the culture medium, the culture medium was changed to fresh medium containing topotecan (1 µM), irinotecan (1 µM), and Dox (1 µM), and the cells were incubated at 37 °C for 24 h. After incubation, the cells were washed with 1× PBS, trypsinized, exposed to 0.2% trypan blue at a 1:1 ratio, and then counted manually using a Neubauer chamber under a microscope. To ensure the reliability of the trypan blue cell exclusion assay results, all experimental conditions were repeated in a MTT assay as described previously with minor modifications [35]. Briefly, 2 × 10^2^ (untreated, DEHP-exposed, and DoxR) cells were seeded in 96-well plates and incubated overnight at 37 °C. After incubation, the cells were treated with tariquidar (1.5 µM) for 2 h in the culture medium (100 μL), the culture medium was changed to fresh medium containing topotecan (1 µM), irinotecan (1 µM), and Dox (1 µM), and the cells were incubated at 37 °C for 24 h. The medium was replaced with 100 μL medium containing 0.5 mg/mL MTT reagent and incubated for 3 h at 37 °C in the dark. The medium was removed, and 100 μL DMSO was added and incubated at 37 °C in the dark. The results were recorded using a microplate reader (BioTek-800^TS^, BioTek Instruments Inc., Winooski, VT, USA) and analyzed using Sigmaplot (Version 12.3, Systat Software Inc., Dusseldorf, Germany).

For western blotting, 1 × 10^5^ cells were seeded in 6-well plates overnight. After incubation, the cells were treated with culture medium containing 1.5 µM tariquidar for 72 h at 37 °C. After incubation, the cells were processed for protein extraction and western blotting as described above (Section 2.7).

### 2.13. Tariquidar-Induced Drug Accumulation and Efflux Analysis

Approximately 1 × 10^5^ untreated, DEHP-exposed and DoxR MDA-MB-231 cells were seeded in 6-well plates. After incubation, a set of cells was treated with tariquidar (1.5 µM) for 2 h in culture medium followed by treatment with Dox (1 µM) for 2 h (Trq + Dox-ptake 2 h). Another set was treated with only Dox (1 µM) for 2 h (Dox-Uptake 2 h). To evaluate drug efflux, a third set was treated with tariquidar (1.5 µM) for 2 h followed by Dox (1 µM) for 2 h and then, the cells were provided fresh medium and incubated for 1 h (Trq + Dox-Efflux 1 h). A fourth set was treated with Dox (1 µM) for 2 h followed by replacement with fresh medium for 1 h Dox-Efflux 1 h). Then, the cells were trypsinized and processed for fluorescence analysis as described above (Section 2.5).

### 2.14. Confocal Imaging

Approximately 2 × 10^4^ untreated, DEHP-exposed and DoxR MDA-MB-231 cells were seeded in 24-well plates containing a circular cover glass and incubated overnight. After incubation, the cells were treated with only Dox (1 µM) for 2 h, tariquidar (1.5 µM) for 2 h followed by Dox (1 µM) for 2 h and a blank set with no treatment. After treatment and incubation, the cells were washed with 1× PBS and fixed with 4% paraformaldehyde for 20 min. Nuclear staining was performed using DAPI, and the cover glass was mounted on microscope glass slides using mounting medium (Vectashield, Vector Laboratories, CA, USA). Then, confocal imaging was performed using an Olympus FV1000 confocal laser scanning microscope.

### 2.15. Evaluation of Mechanism of Cell Death

A total of 4 × 10^5^ untreated, DEHP-exposed, and DoxR MDA-MB-231 cells were seeded in 6 cm plates and incubated overnight. After incubation, the cells were treated with Dox (2 μM) and tariquidar (1.5 μM) for 2 h followed by Dox (2 μM) for 48 h. After 48 h, the cells were harvested using 1× trypsin, proteins were extracted, and western blotting was performed as described above (Section 2.7). For apoptosis analysis by Annexin V-FITC staining, 8 × 10^4^ untreated, DEHP-exposed, and DoxR MDA-MB-231 cells were seeded in 6 cm plates and incubated overnight. After incubation, the cells were treated with Dox (2 μM) and tariquidar (1.5 μM) for 2 h followed by Dox (2 μM) for 48 h. After 48 h, the cells were harvested using 1× trypsin, washed with 1× PBS followed by staining with Annexin V-FITC for 30 min at 37 °C, and flow cytometry was performed using a BD Accuri C6 flow cytometer. The results were analyzed by BD CFlow^®^ software.

### 2.16. Library Preparation and Transcriptome Sequencing (NGS)

The RNA sequencing samples were sent to Biotools Co. Ltd., Taipei, Taiwan. Briefly, 3 µg RNA per sample was used as input material for the RNA sample preparations. Sequencing libraries were generated using the NEBNext^®^ Ultra™ RNA Library Prep Kit for Illumina^®^ (NEB, Ipswich, MA, USA). mRNA purification was performed using poly-T oligo-attached magnetic beads. NEBNext First Strand Synthesis Reaction Buffer (5×) was used for fragmentation. The cDNA was synthesized, and NEBNext Adaptors with hairpin loop structures were ligated for hybridization to the 3′ ends of DNA fragments. The AMPure XP system (Beckman Coulter, Beverly, MA, USA) was used for library fragment purification to select 150–200 bp cDNA fragments. USER Enzyme (NEB, USA) was used to amplify cDNA at 37 °C for 15 min followed by 5 min at 95 °C prior to PCR; further Phusion High-Fidelity DNA polymerase, Universal PCR primers and Index (X) Primer were used to complete PCR. The PCR products were purified and evaluated for library quality assessment. cBot Cluster Generation System using HiSeq PE Cluster Kit cBot-HS (Illumina) was used for clustering the samples. After cluster generation, the libraries were sequenced using an Illumina HiSeq platform, and 125 bp/150 bp paired-end reads were generated. The sequencing data were processed and analyzed using the following Illumina software programs: gene mapping for analyzing differentially expressed genes (DEGs) was performed using Tophat (v2.0.12); Gene Ontology (GO) enrichment of DEGs was assessed using Goseq and topGO (Release 2.12); KEGG analysis to identify enriched biological pathways was performed using KOBAS (v2.0). Ingenuity pathway analysis (IPA) was used in addition to KEGG analysis for validation and identification of pathway enrichment.

### 2.17. Microwestern Array

The high-throughput microwestern array was performed at the microwestern array core facility of the National Health Research Institute (NHRI), Taiwan, as described previously [36]. Briefly, control, DEHP-exposed, and DoxR MDA-MB-231 cells were trypsinized and washed twice with 1× PBS followed by cell lysis to collect total proteins. The NHRI system is high-throughput, and the entire process requires only 50 nL of sample and 0.2 µL of antibody. A 96-well size gel was printed, which can run 96 different blots, and each western blot can run a protein marker and six samples that are separated according molecular weights to avoid cross-reaction problems. The results are consistent with traditional western blotting.

### 2.18. Statistical Analysis

The data analysis was performed using SigmaPlot version 12.3 (registration no. 775000000, Systat Software Inc., Germany). All data are presented as the mean ± standard deviation (SD). One-way analysis of variance (ANOVA) was used to compare multiple groups. The difference between two groups was evaluated using Student’s *t*-test. A *p* value < 0.05 was considered statistically significant.

## 3. Results

### 3.1. Long-Term DEHP Exposure Induced Acquired Drug Resistance in MDA-MB-231 Cells

To evaluate the effect of long-term DEHP exposure on breast cancer cells, DEHP (100 nM) was added to the culture medium of MDA-MB-231 and MCF7 cells, and the cells were cultured with continuous DEHP exposure for three months. After DEHP exposure, the cells were treated with concentration gradients of Dox. Colony formation assays showed that long-term DEHP exposure induced acquired drug resistance in MDA-MB-231 cells; however, no effect was observed in MCF7 cells. DEHP-exposed MDA-MB-231 cells showed a larger colony area (52%) than non-DEHP-exposed MDA-MB-231 cells (27.9%) after Dox (5 nM) treatment, suggesting approximately 2-fold higher proliferation after DEHP exposure (Figure 1A,B). Similarly, DEHP-exposed cells showed resistance to Dox treatment at a high concentration (10 nM) compared to non-DEHP-exposed cells. The results suggest that long-term DEHP exposure at a low concentration (100 nM) induces acquired drug resistance in MDA-MB-231 cells; however, DEHP did not affect MCF7 cells (Figure 1C,D). Additionally, drug-resistant MDA-MB-231 colonies treated with 10 nM Dox were selected and maintained as clones #1, 2, 3, and 4. Dox-resistant MDA-MB-231 cells (DoxR) were established by challenging MDA-MB-231 cells with 10 nM Dox for 72 h in a colony formation assay. Dox-resistant colonies were selected and maintained as DoxR cells (Appendix A).

### 3.2. Long-Term DEHP Exposure Suppresses ROS Formation in MDA-MB-231 Cells

The effect of long term DEHP exposure on ROS generation was evaluated by treating control and DEHP exposed MDA-MB-231 and MCF7 cells with Dox (2 µM) for 0, 12, 18 and, 24 h. ROS level (O_2_^−^) was evaluated by flow cytometry. The results show, in control MDA-MB-231 cells, ROS increased significantly post Dox exposure for 12 and 18 h respectively. DEHP exposed MDA-MB-231 cells show low ROS (4%) in the control condition compared to MDA-MB-231 cells (11.10%). Surprisingly, Dox treatment showed moderately increased ROS in DEHP exposed MDA-MB-231 cells (6.10%) compared to that of significant increase in the control MDA-MB-231 cells (23.37%) at 12 h (Figure 2A,C) and similar can be observed at all time points. On the contrary, MCF7 cells basal ROS level was increased in DEHP exposed cells. Dox treatment resulted in the increase of ROS levels in both control and DEHP exposed MCF7 cells (Figure 2B,D).

### 3.3. DEHP Exposure Enhances Drug Efflux through ABCB1 and ABCC1 Overexpression

Based on the colony formation results showing that long-term DEHP exposure induced acquired drug resistance, we evaluated the role of membrane transporter proteins in drug efflux and measured the mRNA and protein levels of ABC transporters in control and DEHP-exposed MDA-MB-231 and MCF7 cells. The Dox efflux results from flow cytometry analysis showed that DEHP-exposed MDA-MB-231 cells had a higher Dox efflux ability of approximately 162.6 ± 2.01% (*p* < 0.05) than MDA-MB-231 cells without DEHP exposure (Figure 3A,B). However, in DEHP-exposed MCF-7 cells, Dox efflux was only 66.7 ± 4.35% (*p* < 0.001), which was lower than that of MCF-7 cells without DEHP exposure (Figure 3C,D).

The observation that long-term DEHP exposure enhanced Dox excretion from MDA-MB-231 cells may be due to the regulation of ABC transporters (MDR1/ABCB1 and MRP1/ABCC1), so we evaluated the mRNA and protein levels of these transporters in DEHP-exposed and control MDA-MB-231 cells. The mRNA expression of ABC transporters was evaluated by quantitative RT-PCR. The experimental results showed that the mRNA level of MRP1/ABCC1 did not change (Figure 3E); however, the MDR1/ABCB1 mRNA levels were 2-, 4-, and 3-fold higher in DEHP-exposed clones #1, 2, and 4, respectively (*p* < 0.05), compared to non-DEHP-exposed control cells (Figure 3F). Protein level analysis by western blotting (Figure 3G) showed that MDR1/ABCB1 was upregulated in all DEHP-exposed MDA-MB-231 clones, consistent with the change in mRNA levels. The MRP1/ABCC1 protein level was upregulated in DEHP-exposed clones #1 and #3, and a slight downregulation or no change was observed in clones #2 and #4 compared with the control, which was different from the mRNA level results. Immunofluorescence imaging showed enhanced expression of both ABCB1 and ABCC1 in DEHP exposed MDA-MB-231 cells (Appendix A). The results suggest that MRP1/ABCC1 and MDR1/ABCB1 might play an important role in acquired drug resistance through induced drug efflux. As ABCC1 and ABCB1 upregulation was observed in DEHP-exposed clones, only two of the four clones were selected and used for further studies.

### 3.4. Overcoming DEHP-Induced Acquired Drug Resistance Through Tariquidar Treatment

DEHP-induced acquired drug resistance was evaluated by treating two DEHP-exposed clones (#1 and # 2), control cells, and DoxR MDA-MB-231 cells, with 1 μM Dox for 24 or 48 h. Trypan blue cell viability results showed that DEHP-exposed MDA-MB-231 clones #1 and #2 had higher cell viability than both control and DoxR MDA-MB-231 cells in both the 24- and 48-hour treatment groups (Figure 4A). Zebrafish xenograft assay show increased proliferation of DEHP exposed clone #1 after challenging with Dox (1 μM) whereas the proliferation of control MDA-MB-231 cells reduced significantly (Figure 4B). To overcome DEHP-induced acquired drug resistance, we pretreated the cells with tariquidar (1.5 μM) for 2 h and then evaluated the effect of Dox on cell viability. Pretreatment with tariquidar did not show any cytotoxicity alone; however, the combined treatment with tariquidar followed by Dox enhanced the cytotoxicity of Dox (Figure 4C). DEHP-exposed MDA-MB-231 clones #1 and #2 showed 24% and 15% increased cytotoxicity, respectively, with tariquidar pretreatment and Dox combined treatment compared to Dox treatment alone (*p* < 0.05 and 0.001). Tariquidar pretreatment also increased Dox cytotoxicity in control and DoxR MDA-MB-231 cells; however, no significant effect on DoxR MDA-MB-231 cells was observed. DEHP-induced acquired drug resistance was also evaluated against the topoisomerase inhibitor topotecan (1 μM) and camptothecin derivative irinotecan (1 μM). The results showed that DEHP-exposed clones #1 and #2 are more resistant to both drugs than the control cells without DEHP exposure. Tariquidar pretreatment successfully reversed resistance to topotecan and irinotecan in all cells. In DoxR MDA-MB-231 cells, a mild effect of tariquidar pretreatment was observed, showing a nonsignificant reduction in viability, suggesting a different mechanism of resistance from DEHP-induced acquired resistance (Figure 4D,E).

MTT assay was also performed to confirm and validate the results of the trypan blue cell exclusion assay. Similar to the trypan blue cell exclusion assay results, tariquidar (1.5 μM) treatment showed no cytotoxicity in all cells. Tariquidar (1.5 μM) pretreatment followed by Dox (1 μM) treatment increased Dox cytotoxicity by 18% and 14% compared to Dox (1 μM) treatment alone in DEHP-exposed clones #1 and #2 (Appendix A). Tariquidar (1.5 μM) pretreatment successfully enhanced topotecan (1 μM) and irinotecan (1 μM) cytotoxicity in all cells (Appendix A).

The western blotting results showed that tariquidar treatment inhibited the regulation of both MRP1/ABCC1 and MDR1/ABCB1 in control and DEHP-exposed clones #1 and #2 (Figure 4F). However, no effect was observed on DoxR cells post-tariquidar treatment. The results suggest that tariquidar inhibits both MRP1/ABCC1 and MDR1/ABCB1, resulting in the reversal of DEHP-induced acquired drug resistance.

### 3.5. Tariquidar Induced Dox Accumulation through ABCB1 and ABCC1 Inhibition

To evaluate the effect of tariquidar treatment on Dox accumulation and efflux potential, control cells and DEHP-exposed MDA-MB-231 clones #1 and #2 were pretreated with tariquidar followed by exposure to Dox, and the fluorescence intensity of accumulated Dox was evaluated by a BD Accuri C6 flow cytometer. The results show that tariquidar pretreatment increased Dox accumulation in DEHP-exposed clones #1 and #2 compared to control and DoxR MDA-MB-231 cells consistent with the tariquidar-mediated inhibition of MDR-1/ABCB1 and MRP1/ABCC1 regulation limiting the excretion of Dox from DEHP-exposed cells (Figure 5A). In control, 75% Dox accumulated after 2 h of drug treatment (Dox -Uptake 2 h) and 87% Dox accumulated after tariquidar (1.5 µM) pretreatment followed by Dox treatment for 2 h (Trq + Dox-Uptake 2 h). Dox accumulation after 2 h of Dox treatment followed by refreshing Dox free medium for 1 h (Dox-Efflux 1 h) was 64% whereas tariquidar pretreatment followed by 2 h Dox treatment and refreshing fresh medium for 1 h resulted in 69.67% Dox accumulation (Trq + Dox-Efflux 1 h). This indicates control/parental MDA-MB-231 cells can pump out limited Dox due to low expression of MDR-1/ABCB1 and MRP1/ABCC1. Similarly, Tariquidar pretreatment successfully reduced drug efflux in DEHP exposed clone #1 and 2 overexpressing MDR-1/ABCB1 and MRP1/ABCC1. Dox accumulation was increased from 27.47% (Dox-Efflux 1 h) to 59.68% (Trq + Dox-Efflux 1 h) in clone #1 and from 36.10% (Efflux-1 h) to 56.43% (Trq + Dox-Efflux 1 h) in clone #2. These results suggest tariquidar mediated inhibition of MDR-1/ABCB1 and MRP1/ABCC1 by in DEHP-exposed MDA-MB-231 cells increase Dox accumulation and reducing Dox efflux.

Confocal laser scanning microscopy (CLSM) was used to evaluate Dox accumulation in control and DEHP-exposed clones #1 and #2 with and without tariquidar pretreatment. CLSM images and analysis showed slightly higher Dox accumulation in DEHP-exposed clones #1 and #2 than in control and DoxR MDA-MB-231 cells treated with Dox (1 μM) for 2 h; however, after tariquidar pretreatment, higher Dox accumulation was observed in all groups than under the initial conditions (Figure 5B,C). This finding suggests that tariquidar actively limits Dox efflux by inhibiting MRP1/ABCC1 and MDR1/ABCB1.

### 3.6. Tariquidar Enhances Dox-Mediated ROS Formation and Apoptosis

To investigate the mechanism of cell death after Dox and tariquidar-Dox treatment in the control, DEHP-exposed and DoxR MDA-MB-231 cells, western blotting was performed. Based on results from Figure 2A,B, the regulation of ROS markers was evaluated. The expression of Superoxide dismutase 2 (SOD2), Peroxidase (PRX), and Catalase was low in DEHP exposed clones #1 and #2 compared MDA-MB-231 cells. Dox and tariquidar-Dox treatment significantly upregulated the expression of SOD2 and PRX whereas, expression of catalase was upregulated in tariquidar-Dox treatment but decreased in Dox treatment (Figure 6A). As shown in Figure 6B, the cleavage of Caspase 3 and PARP was clearly increased in the Dox treatment group compared to the control group, which showed no cleavage, indicating that Dox induced apoptosis. Furthermore, tariquidar pretreatment along with Dox treatment upregulated the expression of both cleaved caspase 3 and cleaved PARP. The results indicate ROS induced activation of caspase 3 mediated apoptosis in DEHP exposed MDA-MB-231 cells.

To confirm the results of western blotting, Annexin V-FITC apoptosis analysis was performed for evaluating the effect of tariquidar and Dox treatment on apoptosis. Generally, the Annexin V-FITC apoptosis detection assay is performed using two stains, FITC-conjugated Annexin V and propidium iodide (PI); however, in our study, because the experimental drug Dox has intrinsic fluorescence that overlaps with the fluorescence of PI, we used Annexin V-FITC single staining to detect apoptotic cells. Dox (2 μM, 48 h) treatment induced 94%, 65.6%, 69.9%, and 87.9% Annexin V-FITC-positive cells in control cells, DEHP-exposed clone #1, DEHP-exposed clone #2 and Dox-resistant MDA-MB-231 cells, respectively. However, tariquidar (1.5 μM, 2 h) followed by Dox (2 μM, 48 h) treatment increased the number of Annexin V-FITC-positive cells by 96%, 71.9% 74.9%, and 89.2% in the control cells, DEHP-exposed clone #1, DEHP-exposed clone #2 and Dox-resistant MDA-MB-231 cells, respectively. Overall, the DEHP-exposed clones #1 and #2 were more resistant to Dox treatment than parental/control MDA-MB-231 cells; however, tariquidar pretreatment enhanced Dox cytotoxicity (Figure 6C,D). Nevertheless, after Dox and tariquidar-Dox treatment in parental MDA-MB-231 cells, the level of cleaved caspase 3 was lower than that of DEHP-treated clones #1 and #2 and DoxR cells. According to the cell viability results, parental/control MDA-MB-231 cells were more sensitive to Dox than DEHP-exposed clones #1 and #2 (Figure 4B and Appendix A), indicating the involvement of other death pathways. We also evaluated the regulation of BCL-2-mediated cell death markers, including BAX, BCL2, and apoptosis-inducing factor (AIF) in MDA-MB-231 cells (Figure 6E). Downregulation of the prosurvival marker BCL2 was observed in the Dox- and tariquidar-Dox-treated cells compared with the control (nontreated) MDA-MB-231 cells, while AIF was upregulated and BAX showed moderate upregulation. Taken together, the results show that Dox and tariquidar-Dox treatment induced the caspase 3-mediated cleavage of PARP, resulting in caspase 3-mediated apoptosis in DEHP-exposed clones #1 and #2 and DoxR cells. However, in parental/control MDA-MB-231 cells, Dox and tariquidar-Dox treatment induced AIF-mediated apoptosis, also known as caspase-independent apoptosis.

### 3.7. Long-Term DEHP Exposure Regulates Akt Signaling Activation and Induces Cell Proliferation

NGS was performed to evaluate the mechanism of DEHP-acquired drug resistance in MDA-MB-231 cells. The differentially expressed genes between DEHP-exposed clone #1 and MDA-MB-231 cells were evaluated by plotting log2 (fold change) and q value < 0.05 in a volcano plot. A total of 240 genes were found to be differentially expressed, of which 76 genes were upregulated and 164 genes were downregulated in clone #1 compared to MDA-MB-231 cell (Figure 7A). Differentially expressed genes in DEHP exposed clone #1, #2 and DoxR were compared with control MDA-MB-231. Further, the compared groups were compared with each other (Figure 7B) to find potential significantly enriched candidate genes. Seventy six genes overlapping in group A and B were selected for further study. Gene Set Enrichment Analysis (GSEA) showed enrichment of Akt signaling with positive correlation with DEHP exposed clones (Figure 7C). Kyoto Encyclopedia of Genes and Genome (KEGG) pathway analysis showed enrichment of the PI3K/Akt pathway (hsa3371) with a q value of 0.5 and an enrichment factor of 0.01 (Appendix A). Upregulation of colony-stimulating factor 3 (CSF 3), colony-stimulating factor 1 (CSF1), laminin subunit 3 (LAMC3), integrin subunit β (ITGB1), and cyclic AMP-responsive element-binding protein 3 (CREB3) are responsible for PI3K/Akt activation in DEHP-exposed MDA-MB-231 cells (Appendix A). Next, ingenuity pathway analysis (IPA-Qiagen) was performed using the NGS data. A detailed analysis of disease and function predictions revealed enhanced cellular movement, growth, proliferation, and development of breast cancer cells. However, a decline in the cellular death/apoptosis of breast cancer cells was observed (Figure 7D,E).

### 3.8. Long-Term DEHP Exposure Activates PI3K/Akt Signaling

A high-throughput microwestern array was performed to determine the influence of long-term DEHP exposure on MDA-MB-231 cells. The results show downregulation of BAX, a proapoptotic signaling molecule, following DEHP exposure in MDA-MB-231 cells, indicating a loss of apoptotic signaling. Loss of PTEN in DEHP-exposed clones #1, 2, 3, and 4 compared to DoxR and parental MDA-MB-231 cells was also observed. PTEN is a negative regulator of PI3K/Akt, suggesting that PI3K/Akt was upregulated or activated in DEHP-exposed clones. The expression of GSK3β was also downregulated in DEHP-exposed clones #1, 2, 3, and 4 compared to DoxR and parental MDA-MB-231 cells. GSK3β is a tumor suppressor gene, the activation of which inhibits the growth and proliferation of cancer cells through PI3K/Akt signaling (Figure 8A). The results of microwestern array were validated by performing western blotting. The markers of PI3K/Akt signaling PI3K p85, Phospho-PI3K p85 (Tyr458), Phospho-AKT (Ser473), Phospho-GSK3β (Ser9), and Phospho-S6 (Ser235/236) were upregulated in DEHP-exposed clones. Slight downregulation of AKT was observed in clone #2 and DoxR whereas, GSK3β downregulation was observed in DEHP exposed clones #1 and 2. Microwestern array results showed downregulation of Phospho-AKT (Thr308) in all DEHP exposed clones suggesting activation of PI3K/Akt signaling through phosphorylation of AKT at Serine 473 (Figure 8B). Overall, the results suggest that long-term DEHP exposure activates PI3K/Akt signaling increasing cell growth and proliferation.

## 4. Discussion

DEHP is mostly used in vinyl products to enhance endurance and flexibility, and the overall weight of PVC products may be composed of approximately 1–40% DEHP. With an annual production of more than 2 million tons, DEHP is the most widely utilized phthalate [37]. The extensive production and use of plastic materials have resulted in environmental contamination and increased concern about human exposure. Potential exposure to DEHP occurs through the use of plastic for food packaging, medicinal devices, personal care products, and household plastic essentials [38,39,40,41]. A recent study, reported DEHP accounts for highest dietary phthalate exposure in Chinese population which is as much as three time higher than that of Europe and America [42]. Global phthalate exposure and assessment found DEHP as a main non-dietary exposure phthalate [43]. This summons for further detailed study of continuous DEHP exposure and its effects on human health and disease progression. Nonconjugate bound DEHP easily leaches out, resulting in environmental release and human exposure. Phthalates are considered environmental hormones that mimic estrogen function, interfering with endocrine system functioning.

Some studies have reported that phthalates promote cancer progression but not cancer onset in breast and liver cancer. Ito Y et al. performed an in vivo study with PPARα-null mice and showed that dietary DEHP exposure induced inflammation and proto-oncogene expression [44]. DEHP exposure was found to enhance cell proliferation through Wnt/β-catenin signaling and to promote non-small cell lung carcinoma (NSCLC) [45]. Our previous study showed that short-term and high-dose DEHP exposure (100 μM, 48 h) induced camptothecin (CPT) resistance in MCF7 cells through Wnt/β-catenin-associated epigenetic mutations; however, no effect was observed in MDA-MB-231 cells [46]. Phthalates (DEHP, BBP, and DBP) induce the proliferation of MCF7 cells through estrogenic activity, preventing apoptosis in the presence of 17β-estradiol [47], suggesting that DEHP has estrogenic activity; however, in our study, we found that long-term (>3 months) DEHP exposure had no effect on ER-positive MCF7 cells. Several studies have indicated the estrogen-independent activity of DEHP, consistent with the results of our study, which found that triple-negative MDA-MB-231 cells acquired drug resistance following long-term DEHP exposure (Figure 1 and Appendix A). DEHP mediated the promotion and invasion of MDA-MB-231 cells through MMP2/9 overexpression, which activated the expression of the NF-κB subunit p65 [48]. Another recent study proposed that phthalates induced AhR/HDA6/cMyc transactivation, a novel mechanism in ER-negative MDA-MB-231 cells [49]. A toxicological study on the effects of DEHP exposure on MCF7 and MDA-MB-231 cells showed enhanced proliferative activity in both MCF7 and MDA-MB-231 cells, which suppressed tamoxifen-induced apoptosis irrespective of ER status [50].

Reactive oxygen species (ROS) is a term used for several byproducts of oxygen metabolism such as H_2_O_2_, O_2_^−^, OH^−^, which are formed during normal cellular functions and are hazardous for cellular health [51]. ROS levels are basically controlled by enzymes which convert harmful ROS into water and oxygen. SOD1 and SOD2 convert O_2_^-^ into H_2_O_2_ which is further converted to H_2_O and O_2_ by peroxidase, GPX, and catalase [52]. Elevated ROS levels can be monitored by expression of these ROS neutralizing enzymes. Abnormal ROS levels are considered responsible for cancer development, progression and drug resistance [53,54].

Several studies found close co-relation between phthalate exposure and ROS generation. Short term DEHP exposure at high concentration (3 mM) was found to elevate ROS levels significantly in prostate adenocarcinoma cells. Low concentration of DEHP metabolite MEHP also induced ROS production in 24 h [55]. Another recent study evidenced increased proliferation of tumor cell through activation of TB4 gene and ROS generation resulting in primary cilium formation [56]. Literature suggest DEHP exposure results in ROS generation leading cancer proliferation and development following short term DEHP/phthalate treatment. Interestingly, our results found long term DEHP exposure at low concentration reduced ROS formation in MDA-MB-231 cells where as MCF7 showed elevated ROS levels post long term DEHP exposure (Figure 2A–D). Slight increase in ROS level was observed following Dox treatment in DEHP exposed MDA-MB-231 cells, at the same time Dox treatment significantly induced ROS production control MDA-MB-231 cells. Overall, long term DEHP exposure may protect MDA-MB-231 cells to undergo ROS mediated oxidative stress reducing Dox cytotoxicity leading to induced Dox resistance.

Drug resistance is a major limiting factor for cancer prevention and treatment in patients, resulting in high mortality despite progress in the development of chemotherapeutic drugs. Drug resistance can be categorized as intrinsic or acquired based on the time of resistance development, the former being present before treatment and the latter achieved after treatment [57]. The enhanced pumping of chemotherapeutic drugs out of cells leads to reduced accumulation and is considered a crucial mechanism of drug resistance, both intrinsic and acquired, and depends on the presence of efflux machinery (transmembrane transporters). The ABC transporter superfamily is responsible for drug efflux, among which MDR1/ABCB1, MRP1/ABCC1, and BCRP/ABCG2 are major players [17,58]. With the presence of numerous binding sites, ABCB1 and ABCC1 can bind to and pump out a variety of chemotherapeutic drugs, such as Dox in breast cancer; paclitaxel and olaparib in ovarian cancer; and imatinib in chronic myelogenous leukemia [59,60,61].

The Dox accumulation and efflux analysis results showed enhanced drug efflux activity in MDA-MB-231 cells compared to MCF7 breast cancer cells, indicating acquired drug resistance (Figure 3A–D). Acquired drug resistance through the elevated regulation of ABCB1 after chemotherapy is evidenced in myelogenous leukemia, a hematological malignancy [62]. Similarly, our findings show the overexpression of both MDR1/ABCB1 and MRP1/ABCC1 following long-term DEHP exposure (Figure 3E–G and Appendix A). DEHP and MEHP induced the activation of MDR1/ABCB1, resulting in multidrug resistance in colon cancer [20]. The endocrine disruptor DEHP activated MDR1/ABCB1 gene expression in colon cancer [63]. Overall, DEHP exposure results in multidrug resistance, which can be seen in our study as the DEHP-exposed clones #1 and #2 acquired resistance not only against Dox but also against topotecan and irinotecan (Figure 4A–E). To mimic physiological conditions, we exposed MCF7 and MDA-MB-231 cells to low DEHP concentrations (100 nM) for more than 3 months and found enhanced acquired drug resistance in MDA-MB-231 cells though MRP1/ABCC1 and MDR1/ABCB1 overexpression; however, no effect was observed in MCF7 cells.

To address the problem of acquired drug resistance through the regulation of transmembrane transporters (ABC transporters), several strategies have been considered, such as gene knockout of transporter proteins, the use of microRNA, and the development of inhibitors. A study evaluated the effect of ABCB1 silencing, and verapamil and CBT1 (an ABC transporter inhibitor) showed similar activities under both conditions, with the gene-silenced condition being slightly more effective in overcoming drug resistance in osteosarcoma [64]. This study also showed that the specific inhibition of ABCB1 with CBT1 resulted in enhanced drug accumulation; however, ABCC1 regulation still enabled drug efflux, indicating the need for the development of multitarget inhibitors. To date, research has resulted in the development of as many as 3 generations of ABC transporter inhibitors [65]. The first two generations of inhibitors, including the first-generation inhibitors verapamil, tamoxifen, cyclosporine A, and quinidin, and the second-generation inhibitors elacridar (GF120918), biricodar (VX-719), and valspodar (PSC883), did not improve drug efficacy along with potency and increased toxicity [66]. Tariquidar (XR9576) is a third-generation inhibitor with no side effects and high efficiency, and combined treatment with tariquidar and vinorelbine potently inhibited ABCB1, increasing drug cytotoxicity [23].

A pharmacokinetic study of combination therapy with tariquidar and the chemotherapeutic drugs Dox/docetaxel/vinorelbine resulted in 22% more drug accumulation than single therapy in refractory solid tumors in children [67]. Another study demonstrated the specific inhibition of MDR1/ABCB1, which restored the potency of Dox and vinblastine in ABCB1-expressing NCI/ADR (Res) cells, a drug-resistant variant of MCF7 cells, in a tumor spheroid model [68]. Our results show that tariquidar pretreatment reversed the DEHP-induced acquired drug resistance against Dox, topotecan and irinotecan in DEHP-exposed clones #1 and #2 as well as in control MDA-MB-231 cells (Figure 4C–E and Appendix A). Western blot analysis showed the specific inhibition of both MRP1/ABCC1 and MDR-1/ABCB1 post-tariquidar treatment, consistent with prior reports; however, these results (Figure 4E) suggested dual inhibitory activity, as reported in the case of CBT1 [64]. Tariquidar influences the deposition of imatinib in mouse plasma by the dual inhibition of MDR1 and BCRP/ABCG2 [69], which supports the dual inhibitory activity observed in our results.

A recent study reported that the reduced efflux efficiency of ABCB1 substrate Rh-123 and [^3^H] daunorubicin post-tariquidar (100 and 50 nM) pretreatment retained 80% and 50% Rh-123 after 2 h of efflux, which enhanced the sensitivity of MDR EMT6/AR1.0 murine mammary carcinoma cells to paclitaxel, Dox, vincristine, and etoposide [70]. Another clinical study performed to evaluate the effect of tariquidar treatment reported the complete inhibition of ABCB1 in CD56^+^ peripheral blood mononuclear cells expressing high endogenous levels of MDR1/ABCB1, resulting in reduced efflux of Rh-123 compared to pretreatment levels without tariquidar pretreatment [71]. According to our results, tariquidar pretreatment enhanced overall Dox uptake in the control cells as well as in DEHP-exposed clones #1 and #2 but reduced Dox efflux efficiency compared to control MDA-MB-231 cells, in which induced efflux was observed following tariquidar pretreatment (Figure 5A). The Dox accumulation study using confocal microscopy showed increased drug accumulation in DEHP-exposed clones compared to control MDA-MB-231 cells without tariquidar treatment. However, tariquidar pretreatment significantly enhanced Dox accumulation by 2-fold under the control condition (without DEHP exposure), by 4- and 2-fold in DEHP-exposed clones #1 and #2, respectively, and by 2-fold in DoxR MDA-MB-231 cells (Figure 5B,C). Overall, tariquidar specifically inhibited both MRP1/ABCC1 and MDR1/ABCB1-limited drug efflux and induced drug cytotoxicity in DEHP-exposed MDA-MB-231 cells.

Dox is the most effective therapeutic agent in breast cancer treatment. It is an anthracycline with a variety of principle molecular actions, such as DNA strand intercalation, free radical generation, topoisomerase inhibition, membrane oxidation, and mitochondrial dysfunction [72]. In a recent study, Dox induced apoptosis through oxidative stress and caspase and BCL2 family member processing in MCF7 cells [73]. Apoptosis is a type of programmed cell death, and alterations or abnormalities in apoptosis may cause abnormal cell growth. Moreover, apoptosis regulation is important because it plays a significant role in malignancy treatment [74]. BCL 2 family cysteine proteases, i.e., the caspase family, including both signaling molecules and effectors, and ROS play crucial roles in apoptosis activation and execution [75,76]. Caspase 3 is an important effector protein in apoptosis, and a study performed on MCF7 and MDA-MB-231 cells showed that caspase 3 status is responsible for the genistein response, as MCF7 lacks caspase 3 [77]. An in vivo study of cardiomyocytes showed that Dox induces caspase 3 activation-mediated apoptosis in cardiomyopathy [78].

The role of ROS in activation of canonical caspase cascade mediated apoptosis is very well established and several studies have already reported it. ROS mediated apoptosis may follow intrinsic or extrinsic apoptosis signaling leading to apoptosis mediated cell death. ROS instability leads depolarization of mitochondrial membrane releasing cytochrome C which activates caspase 9 through nucleotide binding of APAF-1 and finally caspase 3 activation [79]. Death receptor mediated activation of caspase 8 and caspase 3 have also been reported under increased ROS production activating extrinsic apoptosis signaling regulated by FAS/TRAIL/FADD [80]. Chen et al. found induced ROS leads to ER stress mediated apoptosis in prostate cancer following Isoalantolactone treatment. They also reported elevated ROS levels inhibit STAT3 signaling [81]. In INS-1 cells high concentration of DEHP exposure for 24 h leads to elevated ROS production and oxidative stress resulting in apoptosis and autophagic cell death [82]. Our results showed reduced expression of SOD2, PRX, and catalase in DEHP exposed clones #1 and #2 compared to control MDA-MB-231 cells at basal level. However, Dox and tariquidar-Dox treatment induced expression of SOD2 and PRX in DEHP exposed clones, indicating increase in ROS following Dox treatment which is further elevated in tariquidar-Dox treatment (Figure 6A). At the same time, Dox treatment activated caspase 3-mediated apoptosis in control cells, DEHP-exposed clones #1 and #2, and DoxR MDA-MB-231 cells. Additionally, tariquidar-Dox treatment upregulated cleaved caspase 3 and cleaved PARP (Figure 6B). ROS mediated intrinsic caspase 3-mediated apoptosis signaling pathway activation was observed in the current study.

The levels of cleaved caspase 3 and cleaved PARP were lower, while the sensitivity to Dox was higher in control/parental MDA-MB-231 cells than in DEHP-exposed clones #1 and #2. As shown in Figure 6C,D, the number of Annexin V-positive cells was higher after Dox treatment in control/parental MDA-MB-231 cells than in DEHP-exposed clones #1 and #2, which suggests the involvement of other death pathways. AIF is a known caspase-independent effector of programmed cell death, a mitochondrial flavoprotein that translocate to the nucleus when released from mitochondria in response to death signals [83]. The nuclear translocation of AIF leads to chromatin condensation and DNA fragmentation, resulting in apoptosis/cell death. Several studies have demonstrated the role of AIF in apoptosis. Yang et al. reported that AIF mediated caspase-independent apoptosis in cisplatin-challenged ovarian cancer cells [84]. Dox is also reported to stimulate apoptosis via AIF activity, specifically PARP-mediated AIF nuclear translocation, stimulating DNA breakage and chromatin condensation [85]. Another study showed the involvement of AIF in Dox-mediated DNA fragmentation in cardiomyoblasts [86]. Similarly, we observed the upregulation of AIF in MDA-MB-231 cells treated with Dox and tariquidar-Dox, representing AIF-mediated apoptosis. The expression of the anti-apoptosis marker BCL2 was decreased; however, increased BAX expression was observed in the treatment groups, indicating that AIF mediated apoptosis (Figure 6E).

The regulation of PI3K signaling is crucial for maintaining cellular and metabolic function in both cancerous and normal cells. Apart from metabolic processes, several cellular functions, such as cell cycle regulation, cell growth, proliferation, migration, and death (apoptosis) are influenced by PI3K signaling regulation [87,88,89,90]. P13K responds to extracellular stimuli and regulates the intracellular signaling cascade involving Akt, NFκB, p53, and mTOR to achieve its role in cellular functions. PI3K is activated via direct recruitment to cell membrane growth factor receptors/adaptors through the binding of adaptor subunit SH domains of PI3K to the phosphotyrosine residues of membrane adaptors. This activation results in the production of phosphatidylinositol-3,4,5 triphosphate (PIP3) via the phosphorylation of phosphatidylinositol-4,5 biphosphate (PIP2) in response to platelet-derived growth factor (PDGF) [91]. PIP3 further recruits and activates Akt by binding with phosphoinositide docking sites at the membrane through its PH domain. PTEN converts PIP3 to PIP2 by the dephosphorylation of PIP3 and acts as an antagonist of the PI3K/Akt pathway. However, PTEN levels are regulated by PI3K-Akt signaling, which influences the transcription of PTEN by NFκB [92].

Several studies have reported that the regulation/activation of the PI3K/Akt pathway following DEHP exposure influences cellular growth, proliferation as well as drug resistance. Enhanced proliferation and metastasis of neuroblastoma cells was observed following DEHP exposure-induced activation of PI3K/Akt/mTOR signaling [93]. Another study demonstrated PI3K-Akt-mTOR activation in DEHP-exposed liver cancer cell proliferation [94]. In a recent study, maternal exposure to DEHP resulted in increased PI3K/Akt/mTOR signaling in the F1 and F2 generations. Activation of PI3K/Akt/mTOR rescued testicular cells from apoptosis and promoted proliferation following direct DEHP exposure [95]. Most of these studies used short-term DEHP exposure which resulted in PI3K/Akt activation.

However, in our study, NGS, KEGG and GSEA analysis suggest activation of the PI3K/Akt signaling pathway (hsa3371) long-term DEHP exposure with low concentrations (Figure 7C and Appendix A). PI3K/Akt activation regulates the expression of prosurvival genes, such as c-Myc and BCL-_XL_ via NFκB and cAMP response element-binding protein (CREB) via Akt, resulting in cell survival and proliferation [96]. Based on KEGG pathway analysis, the upregulation of CSF1, CSF3, LAMC3, ITGB1, and CREB3 was responsible for PI3K/Akt activation in long-term DEHP-exposed MDA-MB-231 cells/clone #1 (Appendix A). Enhanced cellular movement, development, growth, and proliferation were observed in DEHP-exposed MDA-MB-231 cells/clone #1 according to IPA analysis of NGS data (Figure 7D,E). Microwestern and western blotting analysis confirmed activation of PI3K/Akt signaling showing upregulation of PI3K p85, Phospho-PI3K p85 (Tyr458), Phospho-AKT (Ser473), Phospho-GSK3β (Ser9), and Phospho-S6 (Ser235/236) along with slight downregulation of AKT in clone #2 and downregulation GSK3β in DEHP exposed clone #1 and #2. Downregulation of the tumor suppressor genes PTEN and GSK3β, and the proapoptotic marker BAX was observed in DEHP-exposed MDA-MB-231 cells following long-term DEHP exposure, which supports the NGS data (Figure 8A,B).

Apart from cellular functions such as growth, development, and proliferation; PI3K/Akt plays an important role in ABC transporter-mediated chemoresistance in different cancers, namely, prostate cancer, breast cancer, colon cancer, and leukemia [97,98,99,100]. As described earlier, PI3K activation results in the regulation of the downstream molecules Akt and NFκB; several studies have shown that NFκB is involved in ABC transporter (MDR1, MRP1 and BCRP) regulation. The promotor region of the MDR1 gene consists of a CAAT box instead of a TATA box, which interacts with the NFκB transcription factor and regulates the expression of the MDR1 gene [101]. Another recent study demonstrated that PI3K-Akt-NFκB mediated the regulation of MDR1/P-gp expression in multidrug-resistance in adriamycin resistant MCF7 cells [102]. Li Yongjun et al. evaluated the effect of the natural compound resveratrol (Res) on adriamycin (ADR)-resistant leukemia cells overexpressing MRP1/ABCC1, PI3K, and p-Akt. The results showed that resveratrol treatment reversed ADR resistance by inhibiting the activation of MRP1/ABCC1, PI3K, and p-Akt [103]. A tumor xenograft model of CH^R^8-5-resistant KB cells showed that ferulic acid-mediated inhibition of PI3K/Akt/NFκB signaling successfully reversed MDR1-mediated drug resistance [104]. Based on the accumulated evidence, PI3K/Akt activation regulates ABC transporter expression and may be involved in ABC transporter-mediated chemoresistance. Our results clearly show the upregulation of MDR1/ABCB1 and MRP1/ABCC1. NGS, microwestern array, and western blotting results show activation of PI3K/Akt signaling. Overall, the activation of PI3K/Akt may be involved in the upregulation of MDR1/ABCB1 and MRP1/ABCC1 in DEHP-exposed clones #1 and #2.

## 5. Conclusions

In conclusion, long-term and low-concentration DEHP exposure (>3 months, 100 nM) induced acquired multidrug resistance in triple-negative breast cancer (MDA-MB-231) cells. The acquired drug resistance in MDA-MB-231 cells is indicative of the ER-independent activity of DEHP. DEHP mediated reduced ROS generation indicates a protective mechanism against anticancer drug and oxidative stress. DEHP induced the expression of MDR1/ABCB1 and MRP1/ABCC1, resulting in enhanced drug efflux, reduced drug accumulation, and ultimately reduced drug cytotoxicity. Tariquidar treatment reversed DEHP-induced acquired multidrug resistance by specifically inhibiting both MDR1/ABCB1 and MRP1/ABCC1 in drug-resistant DEHP-exposed MDA-MB-231 cells, limiting drug efflux potential. Tariquidar treatment increased cytotoxicity of DOX through regulation of ROS mediated caspase 3 activation and apoptosis in DEHP-exposed and parental MDA-MB-231 cells. However, in parental MDA-MB-231 cells, in addition to caspase 3-mediated apoptosis, AIF-mediated apoptosis was also observed, indicating that the sensitivity of untreated cells to Dox was higher than that of DEHP-exposed MDA-MB-231 cells. NGS analysis revealed the activation of the PI3K/Akt signaling pathway, which might be involved in the enhanced cell survival and proliferation of DEHP-exposed MDA-MB-231 cells. Microwestern array showed downregulation of the proapoptotic marker BAX along with the cancer suppressor genes PTEN and GSK3β; however, western blotting results show upregulation of signaling PI3K p85, Phospho-PI3K p85 (Tyr458), Phospho-AKT (Ser473), Phospho-GSK3β (Ser9), and Phospho-S6 (Ser235/236) were upregulated in DEHP-exposed clones, which supports the finding that PI3K/Akt was activated. We predicted the involvement of PI3K/Akt signaling in the induced regulation of MDR1/ABCB1 and MRP1/ABCC1 and ultimately the induced multidrug resistance in DEHP-exposed MDA-MB-231 cells. In our study, we attempted to mimic the physiological DEHP exposure condition, which shows enhanced acquired multidrug resistance, which may result in a reduced effect of chemotherapy on breast cancer patients; however, the overexpression of multiple ABC transporters may result in multidrug resistance, which remains a challenge and represents an opportunity for the development of more potent inhibitors (Figure 9).

## Figures and Tables

**Figure 1 antioxidants-10-00949-f001:**
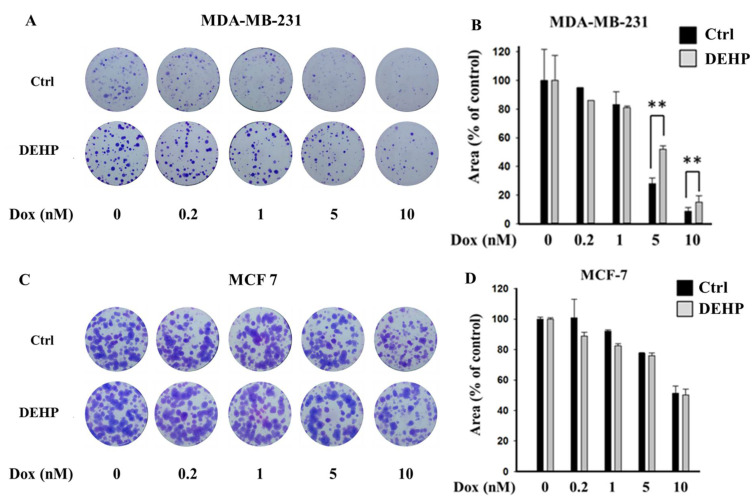
Representative images and quantitative analysis of cell proliferation. (**A**,**B**) Colony formation assay in monolayer cultures of MDA-MB-231; (**C**,**D**) MCF7 cells exposed to DEHP (100 nM) for 3 months and then treated with a concentration gradient of Dox (0.2, 1, 5, and 10 nM) for 72 h and grown for 10 days. ** *p* < 0.001.

**Figure 2 antioxidants-10-00949-f002:**
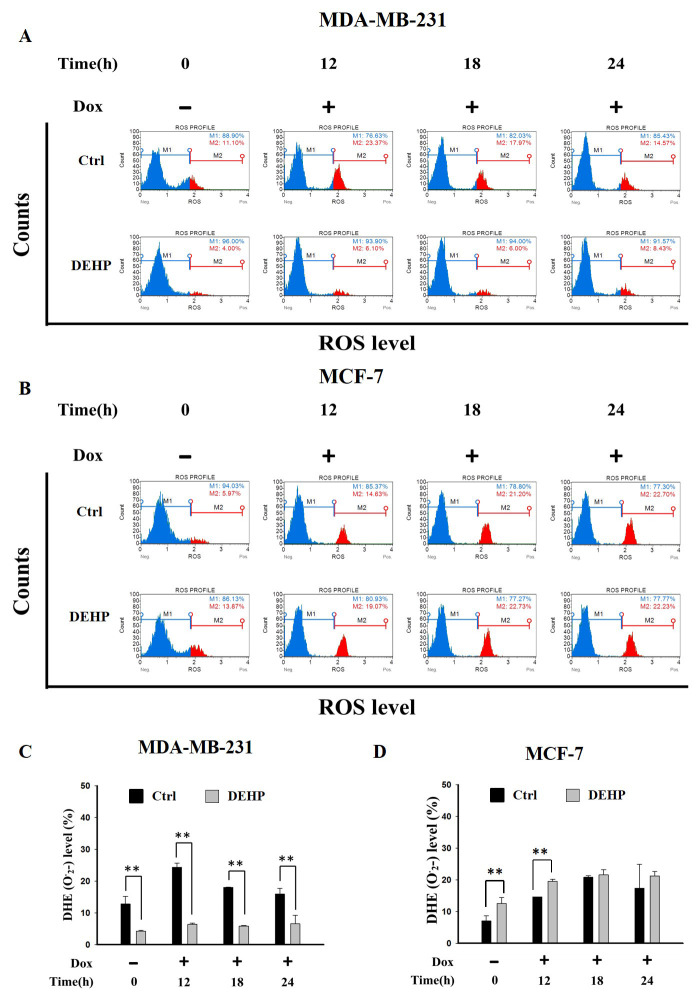
Representative and quantitative analysis of ROS (O_2_^−^) determined by DHE staining and flow cytometry analysis. Control and long term DEHP exposed MDA-MB-231 and MCF7 cells were treated with Dox (2 µM) for 0, 12, 18, and 24 h. Cells were then stained with dihydroethidium (DHE) and analyzed using Guava Muse Cell Analyzer. (**A**,**C**). MDA-MB-231 cells and (**B**,**D**). MCF7 cells. ** *p* < 0.001.

**Figure 3 antioxidants-10-00949-f003:**
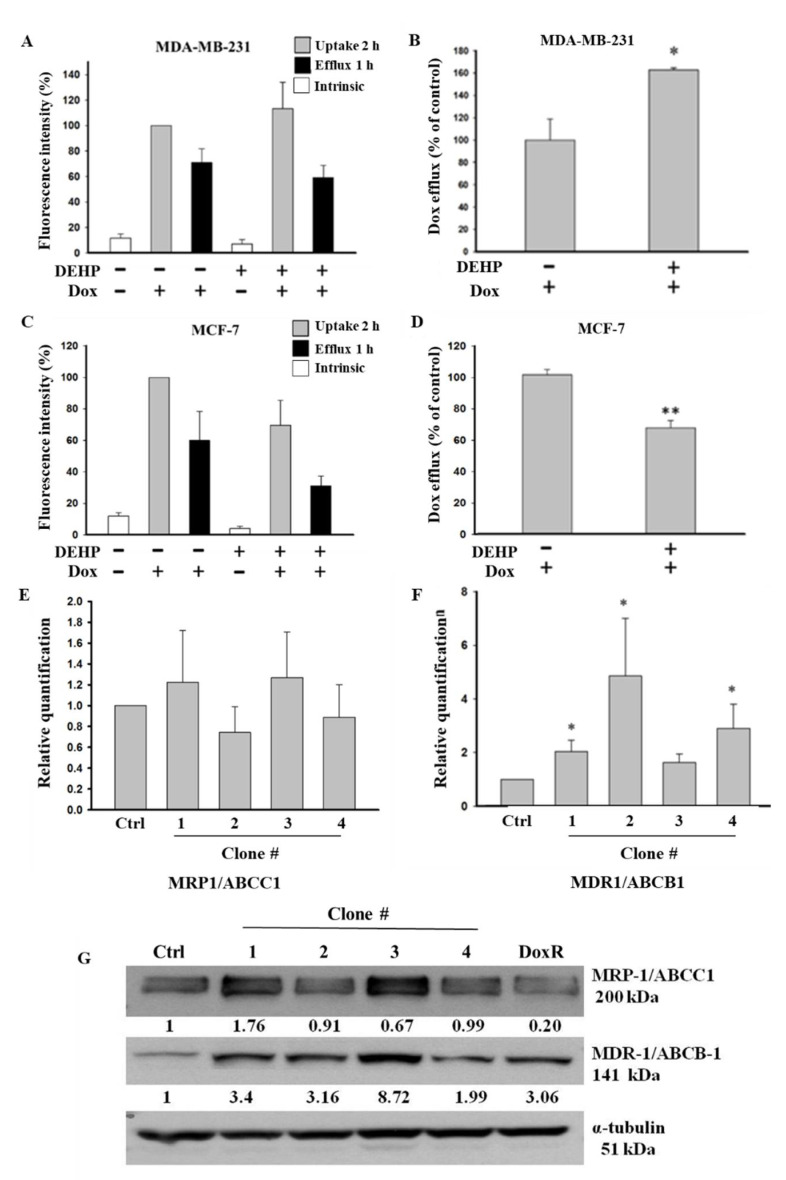
Quantitative results of flow cytometry analysis of drug influx and efflux in MDA-MB-231 and MCF7 cells. (**A**,**B**) Elevated drug influx (original and remaining) and efflux (original quantity minus remaining) were observed in DEHP-exposed MDA-MB-231 cells compared with parental MDA-MB-231 cells. (**C**,**D**) Decreased drug influx and efflux were observed in MCF-7 cells (parental vs. DEHP-exposed). (**E**,**F**) Quantitative analysis of mRNA levels showed no significant change in MRP1/ABCC1 and high MDR1/ABCB1 expression. (**G**) Representative western blot results show the overexpression of MRP1/ABCC-1 and MDR1/ABCB-1 in the long-term DEHP-exposed clones compared to the control and DoxR MDA-MB-231 cells. * *p* < 0.05, ** *p* < 0.001. ■ represent Dox uptake for 2 h, ■ represent Dox efflux for 1 h, □ represent intrinsic/basal fluorescence considered as background.

**Figure 4 antioxidants-10-00949-f004:**
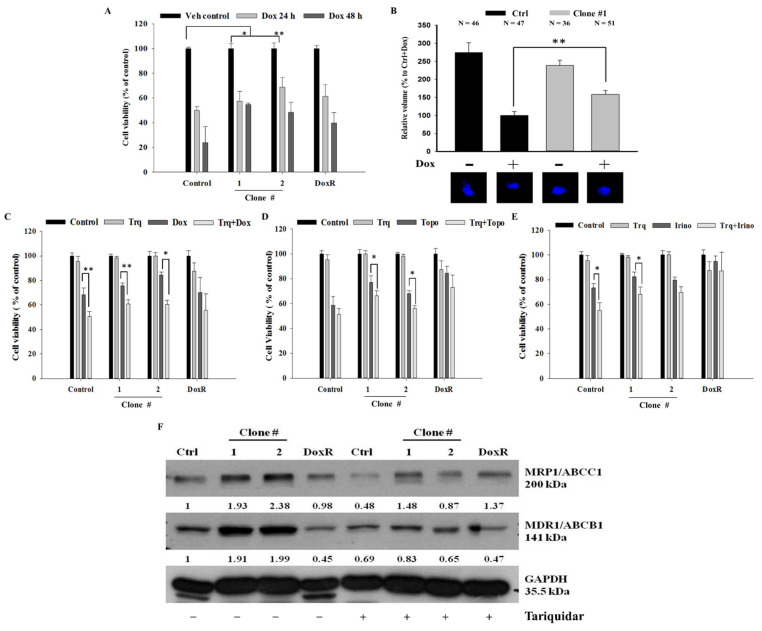
Quantitative analysis of cell viability by Trypan blue cell viability assay. (**A**) Control, DoxR and DEHP-exposed MDA-MB-231 cells (clone #1 & #2) treated with Dox (1 μM) for 24 or 48 h show induced drug resistance. (**B**) Representative and quantitative results of in vivo zebrafish xenograft assay; DEHP induced Dox resistance and increased proliferation is observed in MDA-MB-231 cells injected in 48 hpf zebrafish embryo challenged with Dox (25 μM) for 48 h (N = number of embryos/treatment). (**C**–**E**) Induced cell viability observed in DEHP-exposed clones challenged with Dox (1 μM), topotecan (1 μM), and irinotecan (1 μM) for 24 h; 2-hour tariquidar (P-gp inhibitor) pretreatment enhanced the drug cytotoxicity of Dox, topotecan, and irinotecan, reversing the acquired drug resistance in long-term DEHP-exposed clones and parental MDA-MB-231 cells. (**F**) Western blotting results show the specific inhibition of both MRP1/ABCC1 and MDR1/ABCB1 after tariquidar treatment for 72 h in both control and DEHP-exposed MDA-MB-231 clones (#1 and #2). * *p* < 0.05, ** *p* < 0.001.

**Figure 5 antioxidants-10-00949-f005:**
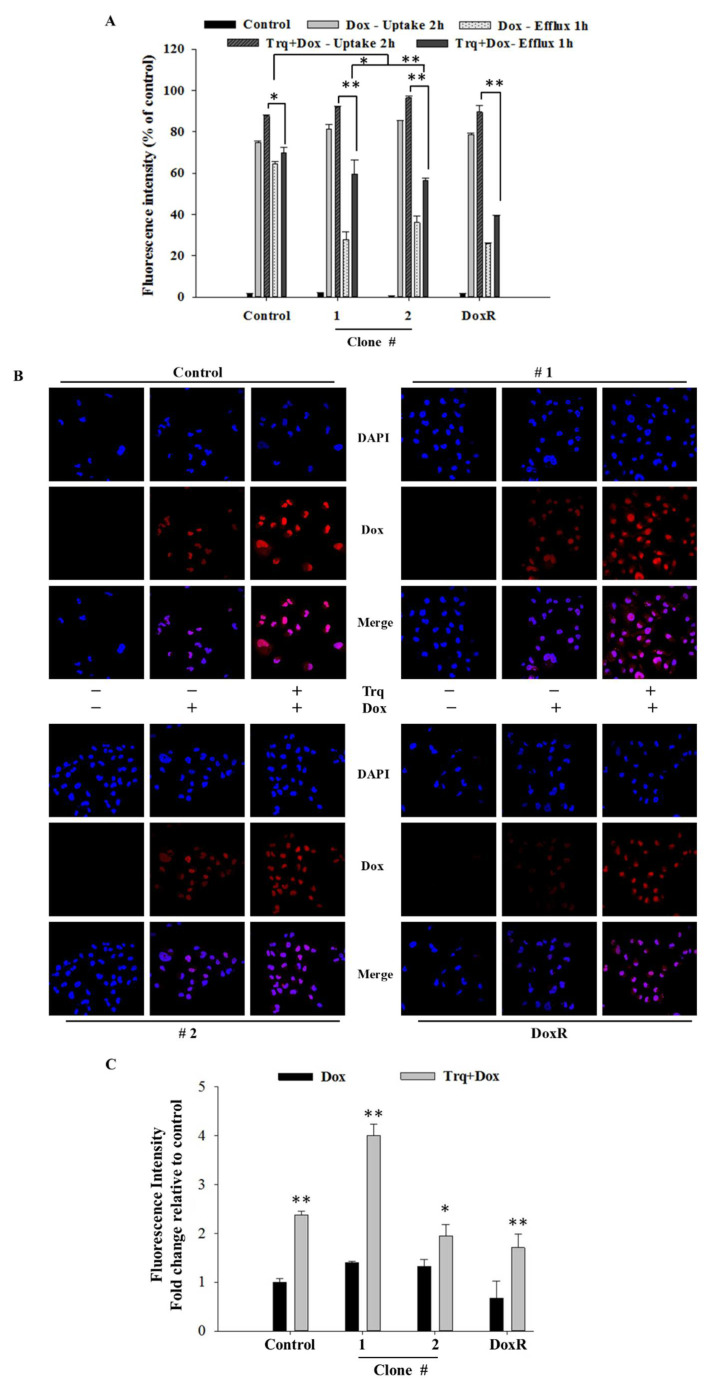
Quantitative and representative results of drug accumulation by flow cytometry and confocal microscopy. (**A**) Enhanced Dox (1 μM) accumulation was observed following 2 h of tariquidar (1.5 μM) pretreatment in DEHP-exposed MDA-MB-231 clones #1 and #2, analyzed by a BD Accuri ^TW^ C6 flow cytometer. (**B**,**C**) Enhanced Dox accumulation (red fluorescence) was observed following 2 h of tariquidar pretreatment along with Dox treatment for 2 h in control cells, DEHP-exposed clones #1 and #2, and DoxR MDA-MB-231 cells. Red fluorescence, Dox; blue fluorescence, DAPI (nuclear staining) as observed using an Olympus FV1000 confocal laser scanning microscope. ** *p* < 0.001, * *p* < 0.05.

**Figure 6 antioxidants-10-00949-f006:**
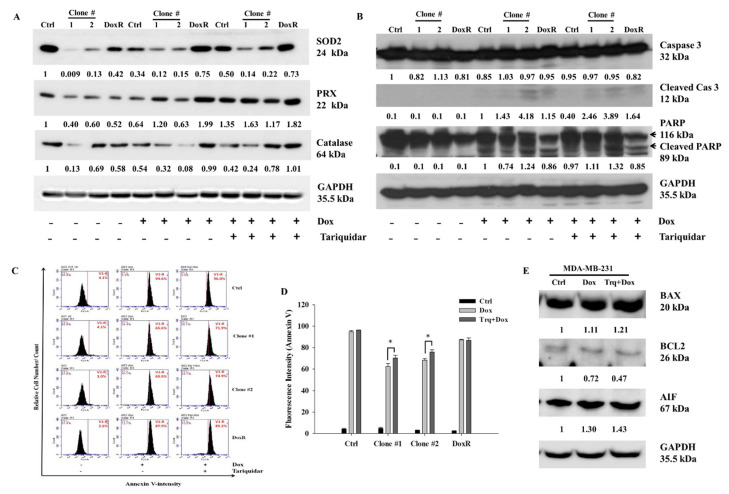
Representative and quantitative results of cell death mechanism in Dox and Tariquidar + Dox treated control and DEHP exposed MDA-MB-231 cells. (**A**,**B**) Western blot results show the upregulation of ROS markers, SOD2, PRX, and catalase along with apoptosis markers cleaved caspase 3 and cleaved PARP by Dox and tariquidar-Dox treatment (2 μM, 48 h) in control and DEHP-exposed clones #1 and #2. (**C**,**D**) Representative and quantitative results of the Annexin V-FITC apoptosis assay. A high number of Annexin V-positive cells was observed after tariquidar (1.5 μM, 2 h) pretreatment plus Dox (2 µM, 48 h) treatment compared to Dox (2 µM, 48 h) treatment alone in DEHP-exposed clones #1 and #2, as evidenced by BD Accuri^TM^C6 flow cytometry. (**E**) In parental/control MDA-MB-231 cells, Dox and tariquidar-Dox treatment (2 μM, 48 h) upregulated BAX and AIF; however, BCL2 was downregulated. The fold change in protein expression from western blots was analyzed by ImageJ 1.42q, NIH, USA. * *p* < 0.05.

**Figure 7 antioxidants-10-00949-f007:**
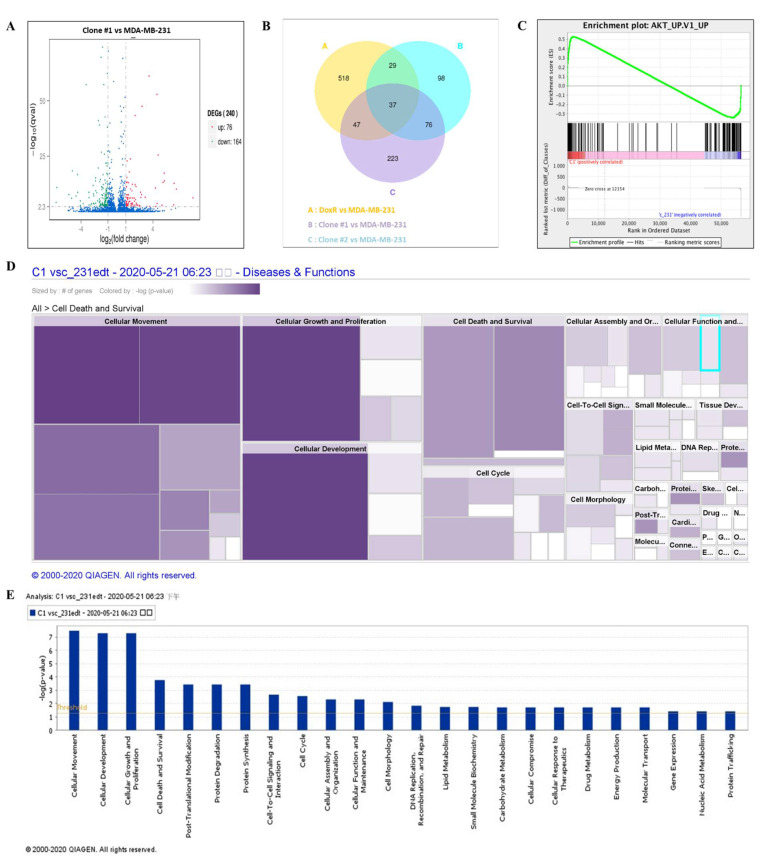
Differential gene expression analysis and the potential mechanism of DEHP-induced drug resistance. (**A**) Volcano plot of the differentially expressed genes between DEHP-exposed clone #1 and MDA-MB-231 cells plotted by log2 (fold change) with q value < 0.05. Significantly differentially expressed genes are represented by ⬤ red dots (upregulation) and ⬤ green dots (downregulation). The x-axis represents the genes expressed in the samples, and the y-axis is the significance of the degree of expression. (**B**) The Venn Diagram of Differential Gene Expression, it presents the number of differentially expressed genes in each group and the overlapping genes between different groups. Group A: DoxR vs. MDA-MB-231 (yellow), Group B: Clone #1 vs. MDA-MB-231 (blue), and Group C: Clone #2 vs. MDA-MB-231 (violet). (**C**) Gene set enrichment analysis showing enrichment plot of Akt signaling with positive correlation of clone #1. Ingenuity pathway analysis (IPA-Qiagen) of the NGS data to compare the disease and molecular function of DEHP-exposed clone #1 with those of parental MDA-MB-231 cells. (**D**) Heatmap analysis of the top 20 cellular diseases and functions of the DEGs involved. Size represents the number of DEGs, and color by –log (*p*-value) represents the associated log of the calculated *p*-value. The color intensity (purple color) of the heatmap squares is proportional to the significance of the *p*-value. A smaller *p*-value (means a larger –log of that value) indicates a more significant association of genes. The *p*-value was calculated using right-tailed Fisher’s exact test. (**E**) Bar graph analysis of the top 20 cellular and molecular functions of the DEGs involved. The X-axis represents cellular or molecular functions, and the Y-axis represents –log (*p*-value). The yellow line represents the threshold.

**Figure 8 antioxidants-10-00949-f008:**
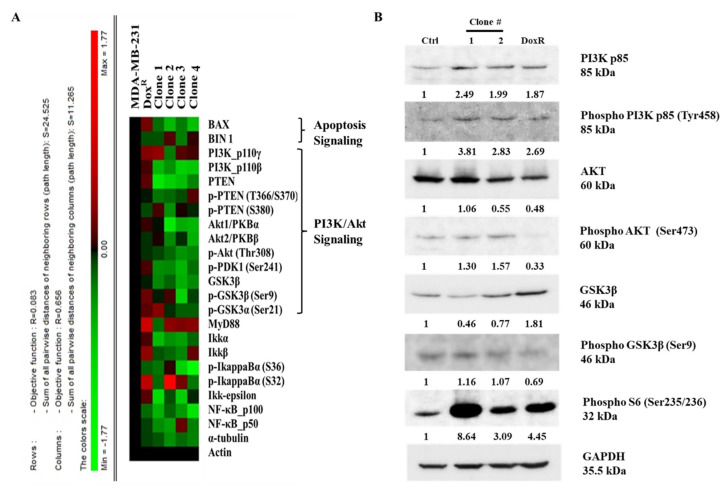
Representative results of the microwestern array and western blotting. (**A**) Target proteins of different signaling pathways, such as apoptosis, PI3K/Akt, and NFκB, were evaluated in control/parental MDA-MB-231, DoxR, and DEHP-exposed clones #1, 2, 3, and 4. Downregulation of PTEN, GSK3β, and BAX was observed in DEHP-exposed clones. The expression levels of different genes are represented by different colors from the control. Green represents downregulation with a minimum score of -1.17, and red represents upregulation with a maximum score of +1.17. (**B**) Representative western blot results show activation of PI3K/Akt signaling in DEHP exposed MDA-MB-231 cells. Upregulation of PI3K p85, Phospho-PI3K p85 (Tyr458), Phospho-AKT (Ser473), Phospho-GSK3β (Ser9), and Phospho-S6 (Ser235/236) was observed along with slight downregulation of AKT in clone #2 and downregulation of GSK3β in DEHP exposed clone #1 and #2.

**Figure 9 antioxidants-10-00949-f009:**
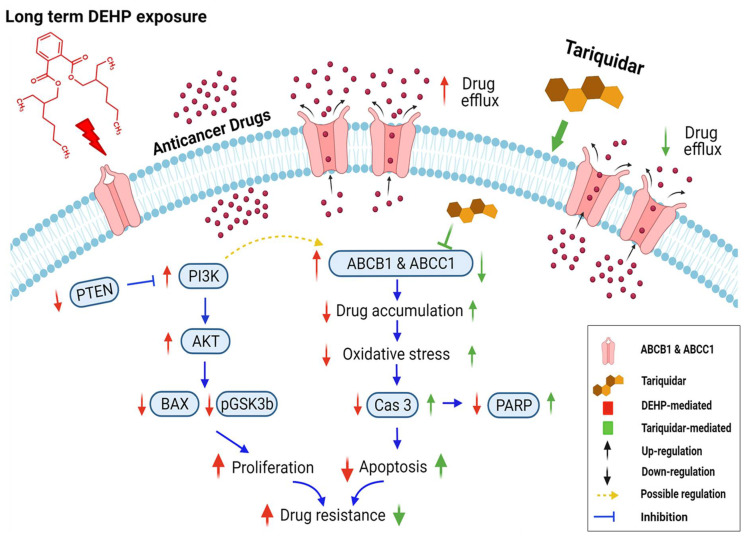
Schematic representation of DEHP-induced acquired drug resistance through MDR1/ABCB1 and MRP1/ABCC1 upregulation in triple-negative breast cancer. Long-term DEHP exposure upregulates MDR1/ABCB1 and MRP1/ABCC1 membrane transporter proteins, increasing drug efflux and decreasing drug accumulation in cells, resulting in acquired multidrug resistance. Long-term DEHP exposure also activates PI3K/Akt signaling, increasing cell survival and proliferation, and it may be involved in the upregulation of ABCB1 and ABCC1. In contrast, tariquidar treatment inhibits the expression of MDR1/ABCB1 and MRP1/ABCC1, decreasing drug efflux, increasing drug accumulation and reversing DEHP-induced acquired multidrug resistance in ER-negative breast cancer cells. **→** indicate DEHP exposure mediated effect, **→** indicate tariquidar treatment mediated effect, **→** indicated interdependent signaling cascade. Created with BioRender.com.

## Data Availability

The authors confirm that the data supporting the findings of this study are available within the article.

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
