# Peer review of "The Long-Term DEHP Exposure Confers Multidrug Resistance of Triple-Negative Breast Cancer Cells through ABC Transporters and Intracellular ROS"

_antioxidants, 2021, doi:10.3390/antiox10060949_

Round 1

Reviewer 1 Report

M​a​n​u​s​c​r​i​p​t​ ​I​D​:​ ​ antioxidants-1239184​

The manuscript entitled “The long-term DEHP exposure confers multidrug resistance of triple-negative breast cancer cells through ABC transporters and intracellular ROS” is written in a clear way. I appreciate the authors' effort in gathering all presented results, however some improvements are needed.

  • Line 33: Please add reference for “physiological concentration of DEHP”
  • Please, justify the concentration of DEHP used in the study
  • Please, justify the concentration of Dox used in the study
  • Please, justify the choice of cell line used in the study: why MDA-MB-231 was compared to MCF-7, not to normal cell line, e.g. MCF-10A or MCF-10F
  • Line 147: the fragment „..maintained under the recommended culture conditions.” should be changed or please, specify “recommended by who?” because ATCC recommended culture medium different than DMEM.
  • Line 191: Please, add information if the measurements was performed in the presence of DEHP (for long term DEHP exposed MDA-MB-231). If not, was the ROS level checked in long term DEHP exposed MDA-MB-231 with and without DEHP?
  • Please specify the passage number of the cells, mainly long term DEPH cells AND please explain what it means “control cell” or ”untreated cells” in the whole manuscript

i.e. line 192 – did “control cells” mean the cells passaged parallelly with “long term exposed cells” – Was the effect of  senescence on test parameters assessed?

  • Line 425 (figure 3A and 3C): Please, add explanation/legend for the first bar (white) – just formality
  • Line 977: some references should be updated, e.g.25, 51

Author Response

Editor                                                                                                                       02, June, 2021

Antioxidants

Dear Editor,

Please find enclosed the revised manuscript entitled “The long-term DEHP exposure confers multidrug resistance of triple-negative breast cancer cells through ABC transporters and
intracellular ROS
” (Manuscript ID: Antioxidants-1239184)” for publication in Antioxidant as an “Original Research Article”. Please see the following pages for the reviewer’s comments and our responses. All revisions and their locations in the manuscript are specified in blue. All authors have read and approved the final manuscript, and there are no known competing interests. Additionally, the revised version of the manuscript was edited by American Journal Experts, a professional English language editing service.

Yours Sincerely,

Dr. Li-Fang Wang
Department of Medicine and Applied Chemistry, Kaohsiung Medical University, Kaohsiung.
Tel: +886-7-3121101 ext. 2217. E-mail: [email protected]

Dr. Chien-Chih Chiu
Department of Biotechnology, Kaohsiung Medical University, Kaohsiung.
Tel: +886-7-3121101 ext. 2368. E-mail: [email protected]

Reviewer 2 Report

The paper is quite interesting, the topic is important and the paper can be considered for publication after few changes. First, the authors should explain because they only considered the DEHP whereas they did not mention the other PAEs, they should stress this information. Further, the could report a section regarding the DEHP determination, how they did it. Figs 2a and 2c are not so readable. Finally, please consider to divide the discussion section in different subsections so it could be easier to read it.

Author Response

(The authors gave the same response as above.)
